# Whole-exome sequencing in UK Biobank reveals rare genetic architecture for depression

Ruoyu Tian[1,22], Tian Ge[2,3,4], Hyeokmoon Kweon[5,6], Daniel B. Rocha[7], Max Lam[4,8,9,10], Jimmy Z. Liu[1,23], Kritika Singh[11,12], Biogen Biobank Team*, Daniel F. Levey[13,14], Joel Gelernter[14,15], Murray B. Stein[16,17,18], Ellen A. Tsai[1], Hailiang Huang[4,19,20], Christopher F. Chabris[6], Todd Lencz[8,9,21], Heiko Runz[1]✉ & Chia-Yen Chen[1]✉

Nearly two hundred common-variant depression risk loci have been identified by genome-wide association studies (GWAS). However, the impact of rare coding variants on depression remains poorly understood. Here, we present whole-exome sequencing analyses of depression with seven different definitions based on survey, questionnaire, and electronic health records in 320,356 UK Biobank participants. We showed that the burden of rare damaging coding variants in loss-of-function intolerant genes is significantly associated with risk of depression with various definitions. We compared the rare and common genetic architecture across depression definitions by genetic correlation and showed different genetic relationships between definitions across common and rare variants. In addition, we demonstrated that the effects of rare damaging coding variant burden and polygenic risk score on depression risk are additive. The gene set burden analyses revealed overlapping rare genetic variant components with developmental disorder, autism, and schizophrenia. Our study provides insights into the contribution of rare coding variants, separately and in conjunction with common variants, on depression with various definitions and their genetic relationships with neurodevelopmental disorders.

Depression is a common and heritable psychiatric disorder with high medical and socioeconomic burden[1,2]. A systematic characterization of the genetic basis of depression may provide novel insights into its etiology and point to novel therapeutic opportunities and patient stratification approaches that may ultimately improve depression treatment. Genome-wide association studies (GWAS) of depression have identified a large number of genetic loci through common variant associations, while the contribution of rare coding variants to depression risk is largely unknown due to the lack of large-scale exome sequenced depression patient samples[3–6]. Recent large-scale exome

sequencing studies have uncovered novel risk genes for neurodevelopmental and psychiatric disorders as well as shared genetic signals between psychiatric disorders[7–11], highlighting the importance of further investigating the impact of rare coding variants on depression through large-scale exome sequencing.

Depression is known to be clinically heterogeneous and therefore, the case samples included in depression genetic studies often show substantial heterogeneity in their phenotypes[12–16]. Previous studies have shown that the common variant genetic architecture varies between different depression definitions[6,15,17]. In particular, it has been

A full list of affiliations appears at the end of the paper.  *A list of authors and their affiliations appears at the end of the paper.
✉e-mail: heiko.runz@gmail.com; chiayenc@gmail.com

shown that the SNP-based heritability ($h^2_g$) can vary widely (0.11–0.33) across different depression definitions in the UK Biobank (UKB), and the genetic correlation ($r_g$) estimates can deviate from one between some of these depression definitions[17]. It is critical to consider this heterogeneity when investigating the impact of rare coding variants on depression.

The recently released whole-exome sequencing data in UKB made it possible to investigate the impact of rare coding variants on depression in the context of its heterogeneity. Here, we analyzed exome sequencing, survey, questionnaire, and electronic health record (EHR) data from 454,787 participants in UKB. As the common variant genetic architecture changes with depression definitions in UKB, we followed the seven previously reported depression definitions with different levels of stringency reported by Cai et al.[17] to identify depression cases and controls in UKB and performed comprehensive genetic analyses on rare coding variants in whole-exome sequencing data across these seven depression phenotypes.

## Results

Following Cai et al.[17], we defined seven depression phenotypes in UKB for our whole-exome sequencing analyses. These depression definitions identify patients who sought medical help for depression from either a general practitioner or a specialist (GPpsy, Psypsy); who had been clinically documented or self-reported as showing symptomatic depression (DepAll, EHR, SelfRepDep); or had one of the two "CIDI" clinical diagnoses based on questionnaire following the Composite International Diagnostic Interview Short Form (CIDI-SF) (lifetimeMDD, MDDRecurr; Supplementary Data 1). We removed participants with self-reported substance abuse, psychotic condition or bipolar disorder from our analysis. We annotated the rare coding variants (minor allele frequency [MAF] <$1.0 \times 10^{-5}$) in UKB exome sequencing data into three categories: protein-truncating variants (PTV), missense variants (further categorized by the MPC deleteriousness score[18]), and synonymous variants. We also stratified genes by pLI (probability of loss-of-function intolerance)[19,20] and used this to classify rare variants. Annotated rare variants were aggregated into 6 groups for rare variants in pLI ≥ 0.9 genes (PTV, MPC > 2, 2 ≥ MPC > 1, 1 ≥ MPC > 0, other missense variants without MPC annotation, and synonymous variants) and 5 groups for pLI < 0.9 genes (PTV, 2 ≥ MPC > 1, 1 ≥ MPC > 0, other missense variants without MPC annotation and synonymous variants; Supplementary Data 2).

We first assessed the impact of exome-wide burden of rare variants on depression risk in unrelated individuals of European (EUR) ancestry (N = 320,356). Exome-wide PTV burden showed significant association with increased risk for GPpsy-, Psypsy-, SelfRepDep- and EHR-defined depression, with the most prominent associations in loss-of-function (LoF) intolerant genes (pLI ≥ 0.9; Fig. 1, Supplementary Fig. 1 and Supplementary Data 3 and 4). In addition, the exome-wide PTV and damaging missense variant (MPC > 2) burden showed strongest effects on EHR-defined depression among all definitions (Fig. 1a; OR = 1.17, 95% CI = 1.13–1.21, $p = 3.57 \times 10^{-18}$ for PTV; OR = 1.08, 95% CI = 1.05–1.23, $p = 8.52 \times 10^{-6}$ for damaging missense variant). No significant association was found for burdens in LoF tolerant genes (pLI < 0.9; Fig. 1b). We down-sampled all depression definitions to the same effective sample size and showed that the overall exome-wide burden association results retained a similar pattern compared with the full sample analysis, suggesting that differential association strengths across definitions were not completely driven by statistical power (Supplementary Fig. 2 and Supplementary Data 5). We repeated the exome-wide burden analysis in UKB participants of South Asian (N = 7053) and African (N = 6290) ancestries, but did not find any significant association, presumably due to limited sample sizes (Supplementary Figs. 3 and 4 and Supplementary Data 6). Finally, we conducted sex-stratified exome-wide burden analyses to identify any sex-specific association[21]. The effects of PTV and damaging missense variant burden were not significantly different

between males and females (Supplementary Fig. 5 and Supplementary Data 7 and 8). Our results demonstrate the overall exome-wide deleterious effects of PTVs and damaging missense variants on depression across definitions.

While previous depression GWAS and our exome-wide analyses showed that both common and rare coding variants contribute to depression risk, we seek to gain more insights into the rare and common genetic architecture and the genetic relationships between different depression definitions. We estimated pairwise burden genetic correlation ($r_g$) based on rare PTV and missense variants by burden heritability regression (BHR)[22]. We found strong $r_g$ between the seven depression definitions based on both rare PTV and missense variants (Fig. 2a, b and Supplementary Data 9). We further compared the relationships between depression definitions by cluster analysis based on rare and common genetic correlations[17]. The clustering patterns were largely consistent across rare PTVs and missense variants: Psypsy-, GPpsy-, SelRepDep-, DepAll- and EHR-defined depression were grouped in a cluster, while the other two most stringent definitions, MDDRecur and lifetimeMDD, were in another cluster (Fig. 2a, b). However, the clustering pattern was different for common variants (from Cai et al.[17]) (Fig. 2c), where EHR-defined depression was least genetically correlated with all the other definitions, although GPpsy-, Psypsy-, and SelfRepDep-defined depression remained in the same cluster as in the rare variant clustering. To assess the level of clustering pattern concordance, we estimated the adjusted Rand Index (ARI) between the clustering of depression definitions based on rare PTV and missense variant and common variant $r_g$. The ARI was 1 (95% CI: 0.52–1.47) between rare PTV and missense variant clustering, as expected for perfect concordance between the two clustering results. The ARI was −0.167 (95% CI: −0.68 to 0.35) between PTV and common variant $r_g$ clusters, which suggests poor concordance between the two clustering results. The ARI between missense variant and common variant $r_g$ clusters was similar to that of the PTV and common variant $r_g$ clusters (ARI = −0.167; 95% CI: −0.68 to 0.35). Overall, these results suggest that the genetic relationships between these depression definitions are different across allele frequency spectrum.

Previous studies showed that genetic prediction using polygenic risk score (PRS) can be further strengthened by incorporating rare mutations with strong effects[23]. Here, we examined the relative contribution of common variant PRS and rare coding variant burden to genetic risk prediction for depression. We performed a meta-analysis ($N_{cases}$ = 157,304, $N_{controls}$ = 576,282) of the depression GWAS from the Psychiatric Genomics Consortium (PGC)[4], Million Veteran Program (MVP)[6], and FinnGen[24] (Release 6) to maximize the prediction power of the depression polygenic score (Supplementary Data 10). We then calculated a PRS for each UKB participant using PRS-CS[25] and the 1000 Genomes EUR samples as the reference panel, and classified each individual by their carrier status of a PTV or damaging missense variant across the exome. For EHR-defined depression (Fig. 3a), in both carriers and non-carriers of damaging rare variants, the prevalence of depression increased with higher PRS, while given the same polygenic risk, carriers of damaging rare variants had increased risk of depression relative to non-carriers. To quantify the relative contributions of common and rare genetic components to depression, we fitted a joint logistic regression to PRS and the carrier status of PTV and damaging missense variants. For EHR-defined depression, common variant PRS (OR per SD change in PRS = 1.34, 95% CI = 1.31–1.37, $p = 4.77 \times 10^{-184}$), PTV (OR per risk allele = 1.16, 95% CI = 1.12–1.21, $p = 4.26 \times 10^{-17}$) and damaging missense variants (OR per risk allele = 1.07, 95% CI = 1.04–1.11, $p = 6.26 \times 10^{-5}$) explained 2.51%, 0.22% and 0.06% of the total phenotypic variation on the liability scale[26], respectively (Supplementary Data 11). Notably, PRS explained 9-fold greater variance than rare variants for EHR-defined depression. We also note that the improvement of genetic prediction by combining PRS and rare coding variant burden is phenotype-dependent (Fig. 3, Supplementary Fig. 6 and Supplementary Data 11). While the genetic prediction can be

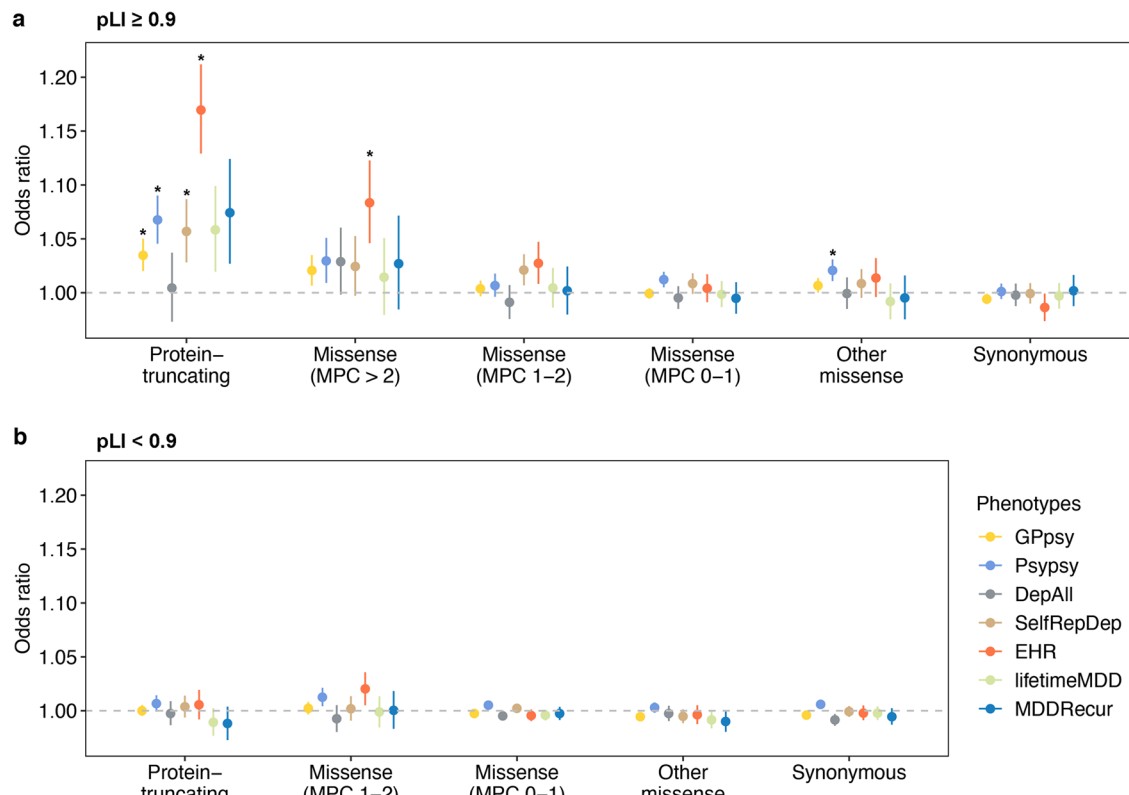

**Fig. 1 | The association between exome-wide rare coding variant burdens and depression with seven different definitions.** *Y*-axis is the odds ratio (OR) of the association between rare variant burden and depression risk. Protein-coding genes were stratified by gene Loss-of-Function (LoF) intolerant with pLI score into (**a**) pLI ≥ 0.9 (LoF intolerant) and (**b**) pLI < 0.9 (LoF tolerant). Rare variants were grouped by functional impact from the most to least severe: protein-truncating, missense (MPC > 2, 2 ≥ MPC > 1, 1 ≥ MPC > 0), other missense (missense variants without MPC score annotation) and synonymous variants. Missense variants in genes (pLI < 0.9) were only annotated into two categories, 2 ≥ MPC > 1 and

1 ≥ MPC > 0. The sample size for each depression definition are as follows: GPpsy: $N_{cases}$ = 111,712, $N_{controls}$ = 206,617; Psypsy: $N_{cases}$ = 36,556, $N_{controls}$ = 282,452; DepAll: $N_{cases}$ = 20,547, $N_{controls}$ = 55,746; SelfRepDep: $N_{cases}$ = 20,120, $N_{controls}$ = 226,578; EHR: $N_{cases}$ = 10,449, $N_{controls}$ = 246,719; lifetimeMD: $N_{cases}$ = 15,580, $N_{controls}$ = 43,104; MDDRecur: $N_{cases}$ = 9462, $N_{controls}$ = 43,104. The gray dashed line represents the null (OR = 1). Each point shows the point estimate of OR from logistic regression. Bars show 95% confidence intervals (CI). *Odds ratios with significant *p* based on Bonferroni-adjusted significance threshold $p < 4.20 \times 10^{-4} = 0.05/119$ (two-sided Wald test; Supplementary Data 3).

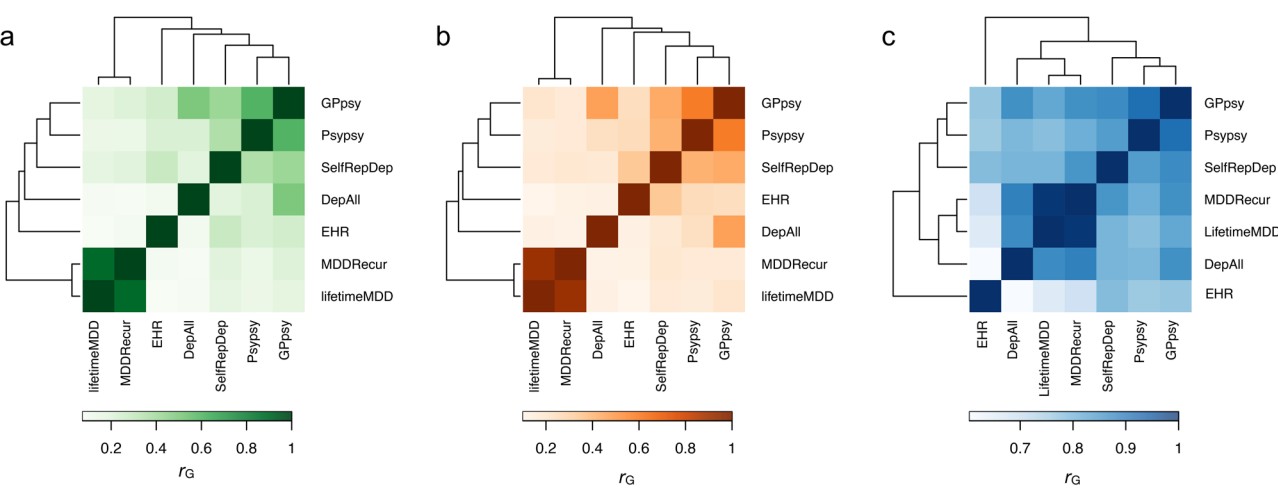

**Fig. 2 | Genetic correlations estimated from rare genetic burden and common variants across depression definitions.** Pairwise burden genetic correlation of depression definitions estimated from (**a**) PTV (MAF < 0.01) and (**b**) missense variants (MAF < 0.01). **c** Pairwise genetic correlations ($r_G$) estimated from common variants between depression definitions (from Cai et al.[17]). All pairwise genetic

correlation estimates were significant at Bonferroni-adjusted threshold ($p < 7.94 \times 10^{-4} = 0.05/63$) based on two-sided Wald tests (Supplementary Data 9), except for the burden genetic correlation from PTV between DepAll and life-timeMDD ($p = 8.7 \times 10^{-3}$) and between DepAll and MDDRecur ($p = 3.77 \times 10^{-3}$).

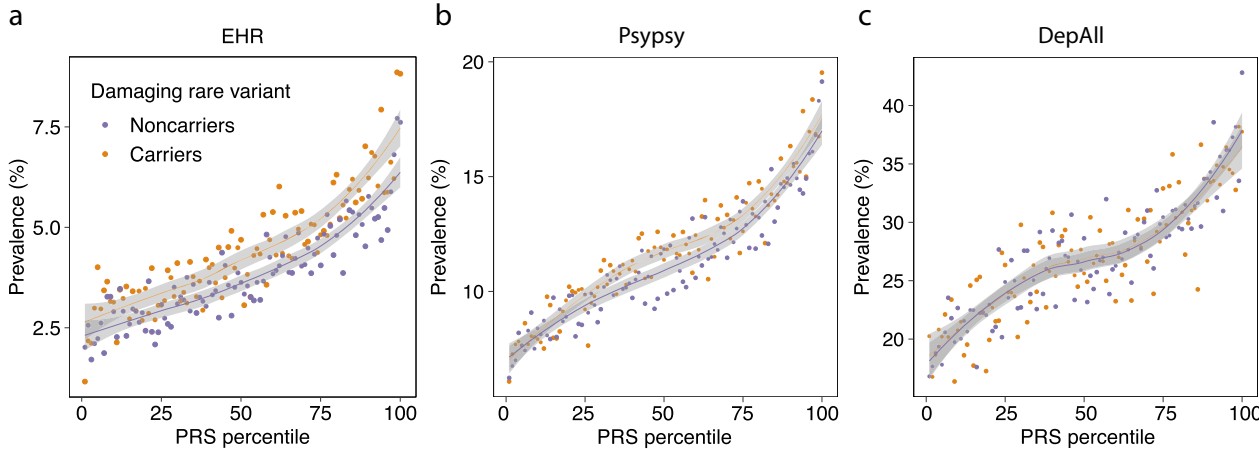

**Fig. 3 | Additive contributions from rare and common variants to depression risk.** The prevalence of (**a**) EHR-, (**b**) Psypsy- and (**c**) DepAll-defined depression against PRS percentile, stratified by exome-wide PTV or damaging missense variant carrier status. The lines represent the locally fitted regression line by LOESS regression, and the gray shading corresponds to the 95% confidence interval of the fitted regression. The sample sizes for each depression definition are as follows: Psypsy: $N_{cases}$ = 36,556, $N_{controls}$ = 282,452; DepAll: $N_{cases}$ = 20,547, $N_{controls}$ = 55,746; EHR: $N_{cases}$ = 10,449, $N_{controls}$ = 246,719.

further improved by incorporating the rare coding variant burden with PRS for EHR-defined depression (likelihood ratio test $p = 3.46 \times 10^{-14}$), the genetic prediction for other depression definitions did not benefit much from incorporating rare coding variant burden (Fig. 3, Supplementary Fig. 6 and Supplementary Data 11). This difference in prediction performance is expected since the effect of rare coding variant burden varies widely across different depression definitions (Fig. 1). Finally, we did not find any significant interaction between damaging coding variant carrier status and PRS for all depression definitions (Fig. 3, Supplementary Fig. 6 and Supplementary Data 11), suggesting additive contributions from PRS and rare variants to depression risk.

To further examine the rare variant genetic overlap between depression and other psychiatric and neurodevelopmental disorders, we performed self-contained gene set burden tests to investigate the impact of rare coding variant burden in genes identified by GWAS or exome sequencing studies of depression related disorders. Rare coding variant burden in depression, schizophrenia or bipolar disorder GWAS genes was not associated with depression (Fig. 4 and Supplementary Data 12 and 13). In contrast, genetic risk derived from exome studies were shared between EHR- and Psypsy-defined depression and psychiatric and neurodevelopmental disorders (Fig. 4 and Supplementary Data 12 and 13), supporting the convergence of genetic risk from rare coding variants in psychiatric and neurodevelopmental disorders.

To gain insights into the biological mechanism underlying rare-variant associations of depression, we performed PTV and damaging missense gene set based burden analyses on Gene Ontology (GO) gene sets[27,28] for biological processes ($N = 7573$), cellular components ($N = 1,001$) and molecular function ($N = 1697$)[29], where these gene sets were implicated in common variant analyses for depression[5,6]. In total, we identified 4 gene sets for EHR-defined depression and one gene set for MDDRecur-defined depression ($p < 0.05/19,757 = 2.53 \times 10^{-6}$) (Supplementary Data 14). We also found significant associations between PTV burden in genes with brain-specific expression (identified in human protein atlas[30]) and EHR-defined depression (OR = 1.26, 95% CI = 1.16–1.36, $p = 1.04 \times 10^{-8}$), where brain-specific PTV burden showed stronger effects than the baseline exome-wide burden (OR = 1.03, 95% CI = 1.01–1.04, $p = 7.19 \times 10^{-5}$) and the PTV burden in genes without tissue specific expression (OR = 1.146, 95% CI = 1.10–1.20, $p = 9.88 \times 10^{-10}$; Supplementary Fig. S7 and Supplementary Data 15 and 16). We also found significant enrichment of PTVs in genes without tissue specific

expression for GPpsy-, Psypsy- and SelfRepDep-depression (Supplementary Data 16).

Next, we aim to examine the human genetic evidence support on FDA approved antidepressants through rare variant gene-set burden analysis. We identified 207 genetic targets of 64 FDA approved antidepressants (e.g., activator, agonist, antagonist, binder, blocker, inhibitor, ligand and modulator; Supplementary Data 17a–c) from the DGIdb browser. We identified one significant rare missense burden association after Bonferroni correction across all tests for Psypsy-defined (i.e., patients who sought medical help for depression from a psychiatry specialist) depression risk (OR = 1.08, 95% CI = 1.03–1.13, $p = 9.83e-4$), with several other suggestive burden associations (Bonferroni-corrected significance per depression definition) for rare PTV and missense variant with MDDRecur-, Psypsy- and EHR-defined depression (Supplementary Data 17d). These findings support the notion that human genetic evidence may enhance depression drug target discovery.

Finally, to discover genes as potential therapeutic targets for depression, we performed gene based PTV and damaging missense burden association tests for all seven depression definitions. We identified two risk genes, *SLC2A1* for EHR-defined depression (OR = 6.01, 95% CI = 3.03–11.94, $p = 2.96 \times 10^{-7}$) and *NOG* (OR = 8.43, 95% CI = 3.50–20.33, $p = 2.03 \times 10^{-6}$) for Psypsy-defined depression through damaging missense variant burden association analysis in UKB whole-exome sequencing data (Supplementary Data 18a, b and Supplementary Fig. 8). Notably, *SLC2A1* is expressed in endothelial cells of the blood-tissue barriers and facilitates transport of glucose into the brain and other tissues[31]. Mutations in *SLC2A1* impair energy supply for the brain and cause GLUT1 deficiency syndrome, characterized by infantile seizures and developmental delay[32–34]. For independent replication, we performed damaging missense burden association tests for *SLC2A1* in two independent biobanks with exome-sequencing and EHR data (Geisinger DiscovEHR cohort and Mass General Brigham Biobank [MGBB]). At current sample sizes, *SLC2A1* burden association was not replicated in the Geisinger DiscovEHR cohort (OR = 1.06, 95% CI = 0.63–1.75, $p = 0.820$; $N_{cases}$ = 21,237, $N_{controls}$ = 45,536), or the Mass General Brigham Biobank (OR = 1.31, 95% CI = 0.53–3.28, $p = 0.562$; $N_{cases}$ = 2,405, $N_{controls}$ = 10,020), although these burden associations showed consistent directions of effect (Supplementary Data 18c). In UKB, the *SLC2A1* damaging missense variant burden also showed consistent direction of effect on all seven depression definitions, albeit the associations were not significant for depression definitions other

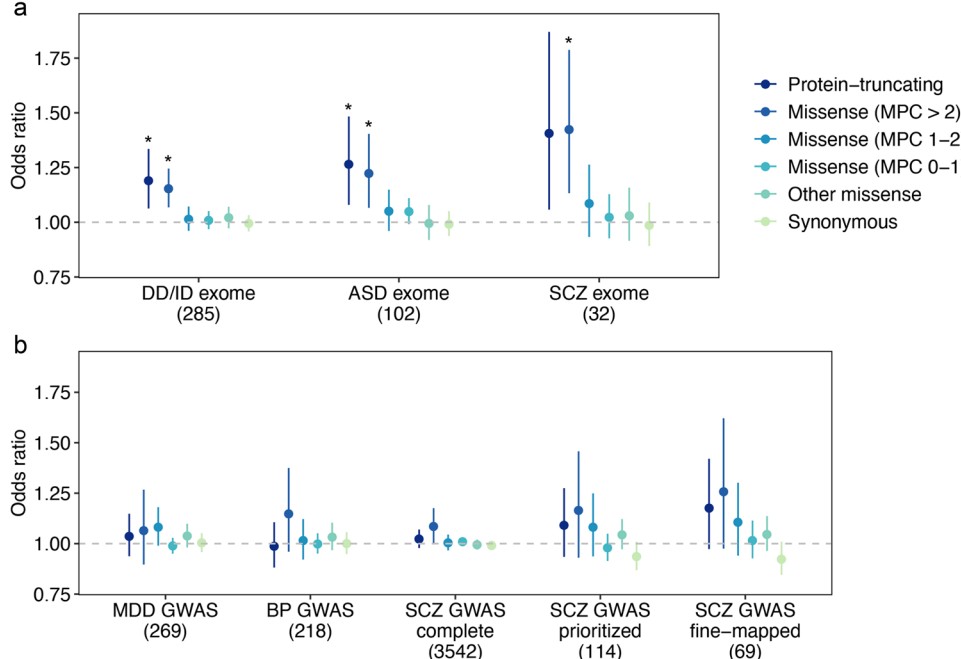

**Fig. 4 | The effects of rare coding variants in psychiatric and neurodevelopmental disease genes on EHR-defined depression. a** The effect of rare variants in psychiatric and neurodevelopmental disease associated genes from previously published exome studies and (**b**) genes from previously published GWAS for psychiatric disorders. We aggregated rare variants of each type (PTV, missense and synonymous) on 8 disease gene sets. From exome studies, we identified 102 autism (ASD) genes (FDR < 0.1)[8], 285 developmental disorder (DD/ID) genes (Bonferroni significant)[9], and 32 schizophrenia (SCZ) genes (FDR < 0.05)[10]. From GWAS, we identified 269 genes for depression (MDD)[5], 218 genes for bipolar disorder (BP)[47], and 3542 complete positionally mapped genes ("SCZ GWAS complete"), 114 prioritized protein-coding genes ("SCZ GWAS prioritized") and 69 fine-mapped genes ("SCZ GWAS fine-mapped") for schizophrenia[48]. *Y*-axis is the odds ratio of the association between rare variant burden for each gene set and depression risk. The gray dashed line represents the null (OR = 1). Each point shows the odds ratio from logistic regression. Bars show 95% confidence intervals. *Odds ratios with significant *p* based on Bonferroni-adjusted significance threshold $p < 1.04 \times 10^{-3} = 0.05/48$ (two-sided Wald test; Supplementary Data 13).

than EHR-defined depression (Supplementary Data 18d). Due to the fact that health survey data were not available in the two replication biobanks, we could not further test the association of *NOG*. While the replication was not successful for both *SLC2A1* and *NOG*, the gene burden associations identified in our study should be viewed as suggestive findings.

## Discussion

With the whole-exome sequencing data from UKB, we presented the largest-to-date whole-exome sequencing study on depression ($N = 320,356$). Collectively, damaging coding variant burden in LoF intolerant genes increased depression risk with various definitions. We showed different genetic relationships between depression definitions across rare and common variants and an additive contribution of rare and common variants to depression risk. We further showed overlapping genetic components between depression and genes identified in exome studies for psychiatric and neurodevelopmental disorders. Through rare coding variant gene set burden analyses, we identified gene sets underlying biological mechanisms/pathways for various depression definitions and demonstrated supportive evidence of gene targets underlying FDA approved antidepressants. Finally, we showed suggestive findings of 2 genes, *SLC2A1* and *NOG*, associated with different depression definitions through gene-based missense burden, but not through PTV burden.

Although we were not able to replicate our findings of suggestive depression genes independent from UKB, our study provides important insights on the rare genetic architecture of depression. This includes analyses that were not previously feasible and comparative analyses of rare and common variants for depression. First, we estimated rare variant genetic correlation between depression definitions using burden heritability regression (BHR), which is a novel method

that can formally estimate genetic correlation based on rare variant burden associations. BHR provided rare variant genetic correlation estimates that can be directly compared with common variant genetic correlations from LD score regression between depression definitions, and we showed that the genetic correlations are different between the same pairs of depression definitions across rare and common variants. This result suggests that the genetic relationship between depression definitions varies depending on the allele frequency spectrum and may be reflecting the complex heterogeneity of depression[15]. Second, we leveraged both whole-exome sequencing and genome-wide genotype data to characterize the relative contributions from exome-wide rare coding variant burden and common variant PRS to depression risk in UKB. The additive contribution of rare and common variants to depression risk is in line with similar findings for other diseases[23,35]. Moreover, we showed that the rare and common variant contributions to depression risk varies with the depression definition, which again highlights the complexity of depression definitions. Third, by leveraging the large sample with whole-exome sequencing data in UKB, we were able to show enrichment of rare variant signals for biological pathways and processes, for example, metabolic process, biosynthetic process, and methyltransferase complex, for depression. Furthermore, we showed enrichment of rare variant burden in the antidepressant target genes for depression. Compared with previous sequencing studies on depression[36], our current study expands on sample size, data quality in terms of sequencing coverage, depression phenotyping, sex (with sex-stratified analyses) and population diversity (with analyses in samples of European, South Asian and African ancestries) and presents a more comprehensive picture of the rare genetic architecture of depression.

Our study is not without limitations. Most notably, the current sample size and depression phenotype definitions in our study did not

fully support gene discovery through rare variants with robust replication. With this, our study emphasizes the need of large-scale, high coverage sequencing data to further study the contribution of rare variants to depression. Our current study analyzed a total of 319,008 discovery samples and 79,198 replication samples of European ancestry with high coverage (20x) whole-exome sequencing data across seven different depression definitions. The lack of replicated gene findings suggests that future rare variant association studies for depression will require advances in one or more of the following directions: (1) Larger whole-exome sequenced samples; (2) Different strategies for sample collection (e.g., diverse population samples); (3) Refined phenotyping and clinical characterization of depression cases, as suggested by the current study where different depression definitions showed different levels of enrichment of rare variant signals; (4) Different types of genome-wide data (e.g., high coverage whole genome sequencing vs whole-exome sequencing); (5) Different, or even novel, rare variant association methodologies, for example, association testing methods incorporating novel variant annotations[37]. We believe that these are important considerations for future studies aiming at depression gene discovery through rare variants.

In summary, through large-scale exome sequencing in UKB, our study examined the impact of rare variants on depression. In conjunction with genome-wide genotyping in UKB and GWAS meta-analysis of depression, we obtained valuable findings into both the rare and common genetic basis of depression. While our study contributed to a more profound comprehension of depression genetics, larger sequencing studies of depression will be necessary for gene discovery through rare variant associations.

## Methods

### The UK Biobank and whole-exome sequencing
The UK Biobank is a large prospective population-based study with over half a million participants recruited across the UK[38]. Phenotypic data collected from each participant includes survey measures, electronic health records, self-reported health information and other biological measurements[39]. The participants have diverse genetic ancestries and overrepresented familial relatedness[39].

Whole-exome sequencing (WES) data from UK Biobank participants was generated by the Regeneron Genetics Center (RGC) as part of a collaboration between AbbVie, Alnylam Pharmaceuticals, AstraZeneca, Biogen, Bristol-Myers Squibb, Pfizer, Regeneron and Takeda. The WES production and quality control (QC) is described in detail in Van Hout et al.[40]. As of November 2020, we obtained QC passed WES data ("*Goldilocks*" set) from 454,787 samples in the UK Biobank.

### Variant annotation
We annotated variants identified in WES by Variant Effect Predictor (VEP) v96[41] with genome build GRCh38. Variants annotated as stop-gained, splice site disruptive and frameshift variants were further assessed using Loss-Of-Function Transcript Effect Estimator (LOFTEE)[20], a VEP plugin. LOFTEE implements a set of filters to remove variants that are unlikely to be disruptive. Those variants labeled as "low-confidence" were filtered out, and we kept variants labeled as "high-confidence". Variants annotated as missense variants were then annotated by MPC score[18], which prioritized damaging missense variants. All predicted variants were mapped to GENCODE[42] canonical transcripts. In total, we identified 649,321 predicted rare PTVs, 5,431,793 missense variants and 3,060,387 synonymous variants with minor allele frequency $<10^{-5}$.

### Phenotyping of depression
Out of all 502,524 UK Biobank samples, we first removed 2256 individuals with self-reported substance abuse (code 1408, 1409 and 1410 in data field 20002), self-reported manic or psychotic condition (code

1291 in data field 20002), bipolar I disorder and bipolar II disorder (code 1,2 in derived data field 20126). We then followed the seven definitions of depression described in Cai et al.[17], including two broad definitions ("GPpsy" and "Psypsy"), a symptom-based definition ("DepAll"), a self-reported definition ("SelfRepDep"), an Electronic health record-based definition ("EHR"), and two CIDI-based definitions (lifetimeMDD" and "MDDRecur"). The former five definitions only required a minimal number of questions to identify depression cases (minimal phenotyping), while the latter two were closer to clinical diagnosis of depression based on Composite International Diagnostic Interview Short Form (CIDI-SF) but were only available for the individuals who participated in the UK Biobank online follow-up.

### Sample filtering and population assignment
We restricted our analyses to 407,139 unrelated individuals and removed 1804 individuals whose reported gender differed from genetic sex or who had sex chromosome aneuploidies. We also removed 133 individuals withdrawn (as of August 24, 2020) from the UK Biobank. To identify UK Biobank samples from different populations for analysis, we performed population assignment based on population structure derived using principal component analysis (PCA) with 1000 Genomes Project (1KG) reference samples ($N_{sample} = 2504$) from 5 major population groups: East Asian (EAS), European (EUR), African (AFR), American (AMR), and South Asian (SAS). We first performed quality control on the 1KG genotype data by retaining only the SNPs on autosomes with minor allele frequency (MAF) > 1% and removed SNPs located in known long-range LD regions (chr6: 25–35 Mb; chr8: 7–13 Mb). We also removed 1 sample from each pair of related samples (greater than second degree) in 1KG. We merged the UK biobank imputed genotype data that was filtered to include imputation quality INFO > 0.8 and MAF > 1% with the 1KG genotype data. We performed LD-pruning at $R^2 = 0.2$ with a 500 kb window. We then computed principal components (PCs) using the LD-pruned SNPs in 1KG sample and derived projected PCs of UK Biobank samples using the SNP-wise PC loadings from 1KG samples. Using the 5 major population labels of 1KG samples as the reference, we trained a random forest model with top 6 PCs to classify UK Biobank samples into 1KG population groups. We assigned UK Biobank samples into one of the 5 populations defined with 1KG reference based on a predicted probability for a specific population group >0.8. We identified 1609 EAS samples, 458,197 EUR samples, 8406 AFR samples, 9224 SAS samples, 1085 AMR samples and 8874 samples without explicit population assignment. Due to the small sample sizes, we did not further analyze samples of EAS and AMR ancestry. We also excluded subjects without an explicit population assignment from our analyses. After initial population assignment, we performed three rounds of within population PCA for AFR, EUR and SAS samples to identify remaining population outliers, each time removing samples with any of the top 10 PCs that was more than 5 SD away from the sample average. We used the in-sample PCs derived after outlier removal in subsequent analyses. We kept individuals with depression case-control status who passed sequencing QC within AFR, EUR and SAS population groups for analysis.

### Exome-wide burden association test
We grouped protein coding genes by pLI (v2.1.1)[19,20] into LoF intolerant (pLI ≥ 0.9) set and LoF tolerant (pLI < 0.9) set. We annotated rare variants by functional consequences into three types, protein-truncating, missense and synonymous. Missense variants were further annotated by MPC score[18] and stratified into 4 groups by predicted deleteriousness: MPC > 2, 2 ≥ MPC > 1, 1 ≥ MPC > 0 and others (referring to missense variants without MPC annotation). In total, we have 17 sets of variants: PTV, MPC > 2, 2 ≥ MPC > 1, 1 ≥ MPC > 0, other missense and synonymous variants in pLI ≥ 0.9 genes (6 groups); PTV, 2 ≥ MPC > 1, 1 ≥ MPC > 0, other missense and synonymous variants in pLI < 0.9

genes (5 groups); PTV, MPC > 2, 2 ≥ MPC > 1, 1 ≥ MPC > 0, other missense and synonymous variants for all genes (6 groups). Note that missense variants on pLI < 0.9 genes were not annotated into MPC > 2 category. Rare alleles of the same variant category on each gene were aggregated into gene-level burden. The summation of the burden on genes in each gene set was the exome-wide burden.

For the exome-wide burden test, we applied logistic regression by fitting exome-wide burden to depression case-control status as the binary response. In the model, we controlled for population structure with top 20 PCs, mean centered age, sex, mean centered age$^2$, mean centered age by sex, mean centered age$^2$ by sex. We performed sensitivity analysis by including the 22 assessment centers as additional categorical covariates in the regression model (Supplementary Data 4). We performed 119 logistic regressions across 7 curated depression definitions and 17 variant sets. We defined a significant threshold $p < 4.20 \times 10^{-4}$ (0.05/119) for the whole-exome burden tests.

### Sex-specific exome-wide burden analyses

To examine the potential sex-specific effect of rare variant burden, we first tested the exome-wide burden association with depression in males and females in a logistic regression controlling for mean centered age, mean centered age$^2$ and top 20 PCs for 11 variant groups (6 groups for LoF intolerant [pLI ≥ 0.9] genes and 5 groups for LoF tolerant [pLI < 0.9] genes) in the EHR-defined depression cohort ($N$ of tests = 33). We also tested the association for protein-truncating and damaging missense variant (MPC > 2) burden in LoF intolerant genes for the other 6 depression definitions ($N$ of tests = 36). Significance threshold was $p < 7.25 \times 10^{-4}$ (0.05/69) for the sex-specific analysis. We further tested if the number of rare variants per sample is different in affected males and affected females, or in control males and in control females. Two-sided Poisson exact test was performed across 7 depression definitions and 2 comparisons (affected female against affected male; control female against male) for PTV and damaging missense variants. In total, there were 28 independent tests and the Bonferroni-corrected significance level was $p < 1.79 \times 10^{-3}$ (0.05/28).

### Rare variant genetic correlation with burden heritability regression (BHR)

To estimate the genetic correlation based on rare variant burdens, we used a recently published method—burden heritability regression (BHR)[22]. Using exome sequencing data in the UK Biobank EUR samples, we performed single variant association tests for all seven depression definitions using the same logistic regression model setting as in the exome-wide burden association tests for all PTVs and missense variants with a minor allele count greater than 5. We estimated rare PTV genetic correlation (including all PTVs with MAF < 0.01) and rare missense variants genetic correlation (including all missense variants with MAF < 0.01) separately following the default settings of BHR with provided baseline model (https://github.com/ajaynadig/bhr). We then performed two-way hierarchical clustering for the seven depression definitions using heatmap.2 function in R v4.2.1 with default settings. For comparison, we extracted common variant genetic correlation for the seven depression definitions from Cai et al.[17] and performed two-way hierarchical clustering based on common variant genetic correlation. We also estimated the adjusted Rand Index (ARI) for pairwise comparison between the rare PTV, rare missense variant, and common variant hierarchical clustering results for the seven depression definitions.

### Polygenic risk score (PRS) analysis

**Meta-analysis.** We meta-analyzed three GWAS of depression: the meta-analysis by Psychiatric Genomics Consortium (PGC)[4] without participants from the UK Biobank or 23andMe; GWAS in Million

Veteran Program (MVP) cohort European sample[6]; and GWAS from FinnGen Release 6. Quality control (QC) pipeline of each set of the summary statistics underwent the following steps if information was available: (1) Remove duplicate and ambiguous SNPs, and SNPs without rsID; (2) Remove SNPs with minor allele frequency (MAF) < 0.01. We used PLINK 1.90 beta[43] to perform an inverse-variance-weighted fixed-effects meta-analysis of the three summary statistics. SNPs appearing in two or more studies were included in the meta-analysis. SNP heritability and LD score regression intercept were computed by LDSC v1.0.1[44]. SNP heritability on the observed scale was transformed to heritability on the liability scale[26], where population prevalence K was set to 0.15. LD score regression intercept was used for evaluating genomic inflation for each study.

**UK Biobank genome-wide genotype data.** Genome-wide genotypes were collected for all UK Biobank participants and imputed using the Haplotype Reference Consortium (HRC)[45] and UK10K + 1000 Genomes[46] reference panels, resulting in a total of more than 90 million variants. We carried out QC steps on the genotyping data by filtering out variants with imputation quality score less than 0.8, or variants with MAF less than 0.01 by PLINK 2.00 alpha[43]. We performed the PRS analysis in EUR samples only, due to limited sample sizes in AFR and SAS populations.

**PRS calculation.** We applied polygenic risk scores-continuous shrinkage (PRS-CS)[25] to estimate the effect sizes of genetic markers. The LD reference panel was precomputed using 1000 Genomes Project phase 3 samples with European ancestry (available at https://github.com/getian107/PRScs). Global shrinkage parameter phi was set to be 0.01 since depression is a highly polygenic trait. PRS of each chromosome for each individual in the validation set was computed by the "--score" function in PLINK 2.00 alpha[43], a linear combination of genotypes weighted by effect size estimates. The final PRS was then summed across chromosome 1 to 22.

**PRS predictive performance evaluation.** To access the predictive performance of PRS, we computed and compared Cox and Snell pseudo $R^2$ for each phenotype with the following the null model (1) and the full model (2):

$$y \sim \beta_0 + \text{covariates} + \varepsilon \qquad (1)$$

$$y \sim \beta_0 + \text{PRS} + \text{covariates} + \varepsilon \qquad (2)$$

where $y$ is the binary response, $\beta_0$ is the intercept, covariates are 20 PCs, mean centered age, sex, mean centered age$^2$, mean centered age by sex and mean centered age$^2$ by sex and $\varepsilon$ is the random error. The partial $R^2$ on the observed scale for PRS was estimated with the same full and null model, which was then transformed to liability scale[26]. Moreover, to compare variance explained by PRS, PTV and damaging missense variants, we also computed Cox and Snell pseudo $R^2$, $R^2$ on the observed scale and the liability scale for PTV and damaging missense variant by replacing the variable PRS with the tested term in the full models. Finally, we tested for the interaction effect between PRS and rare variant carrier status in a logistic regression:

$$y \sim \beta_0 + X_{\text{rare}} + \text{PRS} + \text{PRS} \times X_{\text{rare}} + \text{covariates} + \varepsilon$$

where $X_{\text{rare}}$ is a binary variable denoting an individual carrying a protein-truncating variant or a damaging missense variant. To compare between models, we calculated the ratio between $R^2$ from different models and used likelihood ratio tests for testing the model fit improvement.

## Gene set burden association analyses

We tested the association estimated effect sizes (odds ratios) between depression with different definitions and rare coding variant burdens for specific gene sets, including neuropsychiatric and neurodevelopmental disease genes identified in exome studies[8–10] and GWAS[5,47,48], gene ontology (GO)[27–29], gene expression annotation from the human brain proteome[30], and antidepressant interacting genes[49]. Logistic regression was used to perform the association test between depression and rare coding variant burden, adjusted for top 20 PCs, mean centered age, sex, mean centered age[2], mean centered age by sex and mean centered age[2] by sex.

**Psychiatric and neurodevelopmental diseases genes.** To examine the genetic risk of rare variants in genes identified through common variant associations in GWAS for depression and other psychiatric disorders, we tested 269 depression genes[5], 218 bipolar disorder genes[47] and 3542 positionally mapped genes, 114 prioritized protein-coding genes and 69 fine-mapped genes for schizophrenia[48] identified in GWAS (Supplementary Data 12). We also tested if depression shares rare genetic risk variants with other psychiatric and neurodevelopmental disorders. We identified 102 genes for autism (FDR < 0.1)[8], 285 genes for neurodevelopmental disorder (Bonferroni significant)[9] and 32 genes for schizophrenia (FDR < 0.05)[10]. In total, there were 8 groups of disease associated genes. We used logistic regression to test for association between depression and rare coding variant burdens. The multiple testing was corrected for the number of gene sets ($N = 8$) for each type of rare variants ($N = 6$, including PTV, MPC > 2, $2 \geq$ MPC > 1, $1 \geq$ MPC > 0, other missense variants without MPC annotation and synonymous variants), which leads to a significance threshold of $p < 1.04 \times 10^{-3}$ (0.05/48).

**Gene ontology.** We identified 10,271 gene ontology (GO) gene sets from MSigDB v7.2[29], including biological process ($N = 7573$), cellular component ($N = 1001$) and molecular function ($N = 1697$), which are derived from the Biological Process Ontology by the Gene Ontology Consortium[27,28]. We used Firth's logistic regression[50] to test for association between PTV and damaging missense variant burdens with seven depression definitions, adjusted for top 20 PCs, mean centered age, sex, mean centered age[2], mean centered age by sex and mean centered age[2] by sex. We defined the Bonferroni-adjusted significance threshold as $p < 3.62 \times 10^{-7}$ (0.05/137,948).

**Brain specific expression.** The Human Protein Atlas (HPA)–Brain Atlas[30] integrated 1710 RNA-seq samples across 23 human brain regions from GTEx, cap analysis of gene expression (CAGE) and HPA. In the Brain Atlas, 16,227 genes were kept for analysis after normalization and filtering. Those genes were then categorized by their relative expression in the brain and other tissues: expression elevated in brain (2587 genes), expression elevated in other tissues but expressed in brain (5298 genes) and expression was not tissue specific but expressed in brain (8342 genes; Supplementary Data 15). We applied logistic regression to test for the association of all 6 variant categories (PTV, MPC > 2, $2 \geq$ MPC > 1, $1 \geq$ MPC > 0, other missense variants without MPC annotation and synonymous variants) across the three gene sets. We defined the significance threshold as $p < 2.78 \times 10^{-3}$ (0.05/18).

**Antidepressants interacting genes.** We obtained drug-gene interactions (updated in April 13, 2021) from the DGIdb browser (4.2.0)[49], a database collection of drug-gene interactions and druggable genes from publications and web sources. There are four categories of FDA approved antidepressants in DGIdb, including 21 tricyclic antidepressants (TCAs), 9 selective serotonin antidepressants (SSRIs), 33 serotonin and norepinephrine reuptake inhibitors (SNRIs), and Moclobemide (Supplementary Data 17a). Antidepressant interacting genes were extracted for each drug from the DGIdb browser (version

4.2.0) and there were 207 genes in total, which we defined as the drug-gene interaction gene set (Supplementary Data 17b, c). We used logistic regression to test for associations between 6 types of rare coding variant burdens for the drug-gene interaction gene set with all seven depression definitions (Supplementary Data 17d).

## Gene based burden association analyses

We used Firth's logistic regression to test for associations and estimate effect sizes (odds ratios) between all seven depression definitions and PTV and damaging missense variant (MPC > 2) burdens for all genes across autosomes and chromosome X, with a binary variable denoting rare allele carrying status. We included the same covariates as described above. We excluded genes with less than 10 carriers for PTV or damaging missense burden. In total, 90,738 association tests were conducted and the exome-wide Bonferroni-corrected significance threshold was $p < 5.51 \times 10^{-7}$ (0.05/90,738). Of note, with 2 independent replication samples from Mass General Brigham Biobank (MGBB) and Geisinger DiscovEHR cohort (see below), we performed an inverse-variance-weighted fixed-effects meta-analysis ($N_{cases} = 34,091$, $N_{controls} = 322,338$) of the *SLC2A1* damaging missense burden associations from UKB, MGBB and DiscovEHR.

## Replication of *SLC2A1* burden association in the Mass General Brigham Biobank (MGBB)

**Genotyping data quality control.** The burden of rare missense variants in *SLC2A1* was tested for replication in an independent whole-exome sequencing study in the Mass General Brigham Biobank (MGBB). The MGBB is a hospital-based biobank aiming to collect blood samples, extensive electronic medical records', lifestyle and family history survey data from about 80,000 consented participants as of November, 2021[51]. The currently released 24,787 genotyping samples (as of November, 2021) were sequenced in two batches. The first batch of samples was genotyped with Multi-Ethnic Genotyping Array (MEGA) for 1,416,020 variants. The second batch of samples was genotyped with Expanded Multi-Ethnic Genotyping Array (MEGA Ex) for 1,741,376 variants.

We conducted quality control (QC) for 24,787 genotyped samples from the two batches following a MGBB genotype QC pipeline (https://github.com/Annefeng/PBK-QC-pipeline) by using PLINK1.9, R and Python scripts. We first kept high-quality variants with call rate >0.95 and computed sample-level call rate. We then kept high-quality samples with call rate >0.98 and high-quality variants >0.98. Variant-level missing rate was computed in each batch and variants with missing rate difference >0.75% were filtered out. After merging two genotyping batches, we removed duplicated variants, monomorphic variants and variants not confidently mapped to any chromosomes.

We then combined MGBB genotyping data with the 1000 Genomes (1KG) Project phase 3 data ($N = 2504$)[46], retained overlapping variants, and filtered out variants that were not bi-allelic and strand ambiguous, with minor allele frequency (MAF) < 0.05, or with call rate <0.98. For principal component analysis, we first performed genome-wide LD-pruning at $R^2 = 0.1$ with window size 200 kb and excluded long-range LD regions (chr6:25–35 Mb and chr8:7–13 Mb). Next, we used independent SNPs to compute principal components (PCs) of the merged genotype data. With the 1KG sample label, we used top 6 PCs to train a random forest model and assigned MGBB samples into five populations (prediction probability >0.8), including European (EUR), East Asian (EAS), African (AFR), American (AMR) and South Asian (SAS). We identified 17,287 (69.7%) EUR samples in MGBB.

Within the EUR subset, we removed a total of 513 samples, including samples whose reported sex was different from genetically imputed sex ($F$-statistics < 0.2 were imputed as female; $F$-statistics >0.8 were male); samples with outlying heterozygosity rate (>5 standard deviations from the mean); and related samples (pi-hat > 0.2). After removing variants showing significant ($p < 1.0 \times 10^{-4}$) batch effect, we

performed PCA of QC-ed EUR samples, and removed 73 outlier samples (6 standard deviations away from the mean in top 10 PCs). Finally, 16,701 European samples passed QC and PCs computed with these 16,701 samples were used as covariates to control for population stratification in the replication analysis.

**Whole-exome sequencing data quality control.** The whole-exome sequencing samples were prepared with a custom exome panel (TWIST Biosciences) and sequenced by Illumina NovaSeq with 150 bp paired ends. The sequencing coverage was 20X for more than 85% of exonic targets. Variants were joint called by Genome Analysis ToolKit (GATK) GVCF workflow with HaplotypeCaller in gVCF mode.

We acquired the currently released 26,421 whole-exome sequencing samples and performed the following data quality control with Hail (Hail v0.2; https://github.com/hail-is/hail). We first imported autosomal chromosome VCF files into Hail and merged them into a Hail format matrix table and the following variant-level and sample-level QC were performed by row or column filtering with the matrix table. First, we performed variant-level QC. We split variants with multi-allelic sites into variants with bi-allelic sites. We then retained high-quality variants with genotype quality (GQ) > 20, call rate >0.9, allele count >0, $10 <$ mean depth (DP) $< 200$, allele balance (AB) > 0.9. Then, variants were grouped into SNPs and indels for hard filtering. For SNPs, we kept SNPs with QualByDepth (QD) $\geq 2$, FisherStrand (FS) $\leq 60$, StrandOddsRatio (SOR) $\leq 3$, RMSMappingQuality (MQ) $\geq 40$, MappingQualityRankSumTest (MQRankSum) $\geq -12.5$ and ReadPosRankSumTest (ReadPosRankSum) $\geq -8$. For indels, we kept variants with QD $\geq 2$, ReadPosRankSum $\geq -20$, FS $\leq 200$, and SOR $\leq 10$. In total, there were 10,588,646 variants remaining after QC. Lastly, we conducted sample-level QC and kept 26,421 samples with number of singleton (n.singleton) < 500, genotype quality (GQ) > 40, and call rate > 0.9.

**Phenotyping.** To replicate the *SLC2A1* damaging missense burden association with EHR-defined depression, we identified depression cases and controls following the "EMR-based definition of depression" in Cai et al.[17]. We acquired ICD10 codes for MGBB participants. We first excluded individuals who have any of the following diagnoses: delirium, not induced by alcohol and other psychoactive substances (F05), other mental disorders due to brain damage and dysfunction and to physical disease (F06), personality and behavioral disorders due to brain disease, damage and dysfunction (F07), unspecified organic or symptomatic mental disorder (F09), mental and behavioral disorders due to psychoactive substance use (F10–F19), schizophrenia, schizotypal and delusional disorders (F20–29), manic episodes (F30), and bipolar affective disorder (F31). We then defined depression cases as those individuals who have one or more of the following ICD10 codes: depressive episodes (F32), recurrent depressive disorder (F33), persistent mood (affective) disorders (F34), other mood (affective) disorders (F38), or unspecified mood (affective) disorders (F39); while controls are those individuals who do not have any of the above diagnoses.

**Variant annotation and *SLC2A1* gene based burden analysis.** We annotated variants in *SLC2A1* as described above. In total, we identified 27 damaging missense variants (MPC > 2) with minor allele frequency <0.001. We performed the same regression analysis as in the UKB: we fitted a Firth's logistic regression by regressing depression case-control status on *SLC2A1* damaging missense variant carrier status, and controlled for population stratification with top 20 PCs, mean centered age, sex, mean centered age², mean centered age by sex, mean centered age² by sex.

**Replication of *SLC2A1* burden association in the Geisinger DiscovEHR cohort**
**Sample description.** Applying the same EHR-based phenotype definition, 29,583 cases and 65,599 controls were identified from

the individuals in the Geisinger DiscovEHR cohort[52] who were born before 1990 and whose exome sequencing data were available. We then restricted the sample to individuals of European genetic ancestry on the basis of a principal component analysis (PCA) with 1KG reference samples (see below for the detail). We also removed individuals who did not report to be "White" ethnicity or whose reported sex did not match their genetic sex. We then retained unrelated individuals such that there are no close relatives up to the third-degree of relatedness. As a result, 21,237 cases and 45,536 controls were used in the replication analysis (62.3% female, average age 63.6).

**Identification of the European genetic ancestry group.** We identified individuals of European genetic ancestry by using a PCA with 1KG reference samples. In the 1KG data, we only kept SNPs that have MAF > 0.01 and Hardy–Weinberg equilibrium exact test $p$ value > $10^{-6}$ within each major population group in the 1KG data. We also removed SNPs in long-range LD regions and dropped one individual from each related pair (greater than third degree). We then merged with the 1KG data the DiscovEHR imputed genotype data including only high-quality common bi-allele SNPs (MAF > 0.01, imputation quality INFO > 0.9, average maximum posterior call >0.95, no strand-ambiguity).

We used R package *bigsnpr*[53] to compute 10 PCs in the 1KG sample and projected the PCs to the DiscovEHR data. The PCA procedure implemented in this package first prioritizes variants with clumping by higher MAF ($r^2 < 0.2$) and iteratively removes SNPs in long-range LD regions. The package also implements an optimized procedure to compute projected PCs, which can reduce shrinkage bias that a simple projection from the reference sample may suffer[54]. Using the 1KG data with 10 PCs, we trained a random forest model to classify the individuals in the DiscovEHR into 5 major population groups of the 1KG, where we excluded individuals from Finnish and Iberian populations from the 1KG training data. We assigned European ancestry to individuals if its predicted probability is greater than 0.95.

To further restrict to individuals of homogenous European ancestry and remove poor quality samples, we also filtered out individuals with excess ancestry-adjusted heterozygosity by following the same procedure implemented in Bycroft et al.[39]. This procedure identifies such individuals by finding outliers in PC-corrected heterozygosity and genotype missing rates. Then, we performed a PCA within the remaining individuals and dropped individuals if any of their 10 PCs was more than 5 standard deviations away from the average. We then re-computed PCs in the remaining samples and used these PCs in the replication analysis. In these in-sample PCA procedures, we derived PCs only with the unrelated individuals and projected them to the rest of the samples.

**_SLC2A1_ gene based burden analysis.** The replication was conducted only for the association between the burden of damaging missense variants (MPC > 2) in *SLC2A1* and EHR-defined depression. In total, 67 carriers for damaging missense burden from 32 variants in *SLC2A1* were identified from the analysis sample. Following the same model specification, we fitted a Firth's logistic regression, where we regressed the case-control status on the burden carrying status while controlling for the same set of covariates as described above.

**Reporting summary**
Further information on research design is available in the Nature Portfolio Reporting Summary linked to this article.

## Data availability
The full gene burden association results from UK Biobank in this study can be found at https://doi.org/10.5281/zenodo.10511823. All phenotypic and genotypic data for the UK Biobank are available to

researchers with approved data access from the UK Biobank (https://www.ukbiobank.ac.uk/enable-your-research/register). MGBB data are not publicly available due to privacy and ethical restrictions. Please contact the MGBB for further information on data access (https://www.massgeneralbrigham.org/en/research-and-innovation/participate-in-research/biobank/for-researchers). Please contact the Geisinger DiscovEHR for further information on data access (https://www.geisinger.org/precision-health/mycode/discovehr-project). GWAS summary statistics from FinnGen can be downloaded at https://www.finngen.fi/en/access_results. Meta-analysis of depression by PGC (excluding UK Biobank and 23andme participants) can be downloaded at https://www.med.unc.edu/pgc/download-results/mdd/. Summary statistics of GWAS on samples of European ancestry in Million Veteran Program (MVP) was obtained through MVP Project Proposal MVP200097. pLoF Metrics is available at https://storage.googleapis.com/gcp-public-data--gnomad/release/2.1.1/constraint/gnomad.v2.1.1.lof_metrics.by_gene.txt.bgz. The MPC score is available at ftp://ftp.broadinstitute.org/pub/ExAC_release/release1/regional_missense_constraint/. Human protein atlas data is available at https://www.proteinatlas.org/humanproteome/brain/human+brain. Drug gene interaction database is available at https://www.dgidb.org/.

## Code availability

Software used for analysis includes R v4.2.1, Python v3, burden heritability regression v0.5.0-alpha (https://github.com/ajaynadig/bhr), Variant Effect Predictor (VEP) v96 (https://useast.ensembl.org/info/docs/tools/vep/index.html), LOFTEE (https://github.com/konradjk/loftee), PRS-CS v1.0.0 (https://github.com/getian107/PRScs), PLINK1.90b (https://www.cog-genomics.org/plink/), PLINK2.00a (https://www.cog-genomics.org/plink/2.0), Hail v0.2 (https://github.com/hail-is/hail), and LD Score regression v1.0.1 (https://github.com/bulik/ldsc). Analysis codes used in this manuscript can be found at https://doi.org/10.5281/zenodo.10511823.

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

## Acknowledgements

We thank all the participants and researchers of the UK Biobank, FinnGen, Million Veteran Program, Mass General Brigham Biobank, the MyCode Community Health Initiative, and the MyCode Research Team. We thank the members of the Geisinger-Regeneron DiscovEHR Collaboration who have been critical in the generation of the data used in this study. We thank the Million Veteran Program for kindly providing the GWAS summary results. M.L. and T.L. were supported by the National Institute of Mental Health of the National Institutes of Health (NIH) under award no. R01MH117646 (T.L., principal investigator). The content is solely the responsibility of the authors and does not necessarily represent the official views of the NIH. This work was supported in part by the PsycheMERGE Consortium (NIMH, R01MH118233; H.K., D.B.R., C.F.C., and T.G.). We thank Sally John for critically revising the manuscript and the support from the Biogen Biobank Team.

## Author contributions

C.-Y.C. and H.R. conceived and supervised the study. R.T., T.G., H.K., D.B.R., and C.-Y.C. performed the analyses. R.T. wrote the manuscript. T.G., H.K., D.B.R., M.L., J.Z.L., K.S., D.L., J.G., M.B.S., E.A.T., H.H., C.F.C., T.L., H.R., and C.-Y.C. critically revised the paper. All authors reviewed and approved the final version of the manuscript.

## Competing interests

R.T. is an employee of Dewpoint Therapeutics. E.A.T., C.-Y.C., and H.R. are employees of Biogen. J.Z.L. is an employee of GlaxoSmithKline plc. K.S. is an employee of Novartis. M.B.S. has in the past 3 years received consulting income from Acadia Pharmaceuticals, Aptinyx, atai Life Sciences, BigHealth, Biogen, Bionomics, BioXcel Therapeutics, Boehringer Ingelheim, Clexio, Delix Therapeutics, Eisai, EmpowerPharm, Engrail Therapeutics, Janssen, Jazz Pharmaceuticals, NeuroTrauma Sciences, PureTech Health, Sage Therapeutics, Sumitomo Pharma, and Roche/Genentech. M.B.S. has stock options in Oxeia Biopharmaceuticals and EpiVario. He has been paid for his editorial work on Depression and Anxiety (Editor-in-Chief), Biological Psychiatry (Deputy Editor), and UpToDate (Co-Editor-in-Chief for Psychiatry). He has also received research support from NIH, Department of Veterans Affairs, and the Department of Defense. He is on the scientific advisory board for the Brain and Behavior Research Foundation and the Anxiety and Depression Association of America. The remaining authors declare no competing interests.

## Additional information

[1]Biogen Inc, Cambridge, MA, USA. [2]Psychiatric and Neurodevelopmental Genetics Unit, Center for Genomic Medicine, Massachusetts General Hospital, Boston, MA, USA. [3]Department of Psychiatry, Massachusetts General Hospital, Harvard Medical School, Boston, MA, USA. [4]Stanley Center for Psychiatric Research, Broad Institute of MIT and Harvard, Cambridge, MA, USA. [5]Department of Economics, School of Business and Economics, Vrije Universiteit Amsterdam, Amsterdam, The Netherlands. [6]Autism & Developmental Medicine Institute, Geisinger Health System, Lewisburg, PA, USA. [7]Phenomics Analytics and Clinical Data Core, Geisinger Health System, Danville, PA, USA. [8]Division of Psychiatry Research, The Zucker Hillside Hospital, Northwell Health, Glen Oaks, NY, USA. [9]Institute of Behavioral Science, Feinstein Institutes for Medical Research, Manhasset, NY, USA. [10]North Region, Institute of Mental Health, Singapore, Singapore. [11]Division of Genetic Medicine, Department of Medicine, Vanderbilt University Medical Center, Nashville, TN, USA. [12]Vanderbilt Genetics Institute, Vanderbilt University Medical Center, Nashville, TN, USA. [13]Department of Psychiatry, Yale University School of Medicine, New Haven, CT, USA. [14]VA Connecticut Healthcare Center, West Haven, CT, USA. [15]Departments of Psychiatry, Genetics, and Neuroscience, Yale University School of Medicine, New Haven, CT, USA. [16]VA San Diego Healthcare System, San Diego, CA, USA. [17]Department of Psychiatry, University of California San Diego, La Jolla, CA, USA. [18]Herbert Wertheim School of Public Health and Human Longevity Science, University of California San Diego, La Jolla, CA, USA. [19]Analytic and Translational Genetics Unit, Massachusetts General Hospital, Boston, MA, USA. [20]Department of Medicine, Harvard Medical School, Boston, MA, USA. [21]Departments of Psychiatry and Molecular Medicine, Zucker School of Medicine at Hofstra/Northwell, Hempstead, NY, USA. [22]Present address: Dewpoint Therapeutics, Boston, MA, USA. [23]Present address: GlaxoSmithKline, Upper Providence, Philadelphia, PA, USA. ✉e-mail: heiko.runz@gmail.com; chiayenc@gmail.com

## Biogen Biobank Team

Chia-Yen Chen ⓘ [1]✉, Ellen A. Tsai ⓘ [1] & Heiko Runz ⓘ [1]✉

A full list of members and their affiliations appears in the Supplementary Information.

