## [Peer Review File · Nature Communications]

Whole-exome sequencing in UK Biobank reveals rare genetic architecture for depressionEditorial Note: This manuscript has been previously reviewed at another journal that is not operating a transparent peer review scheme. This document only contains reviewer comments and rebuttal letters for versions considered at *Nature Communications* .

REVIEWERS' COMMENTS

Reviewer #1 (Remarks to the Author):

This paper is an exome analysis of different definitions of depression in the UK biobank and makes three points: there is an excess of damaging coding variants in cases of depression, there are different genetic relationships between definitions across common and rare variants, and the effect of damaging coding variant burden and polygenic risk score on depression risk is additive. These are interesting claims and overall the paper does not oversell the findings (as they say, the association results for the rare variant gene burden tests have to be taken as preliminary not definitive) I have two suggestions to improve the manuscript:

- 1) The claim that there are different genetic relationships between definitions across common and rare variants relies on a cluster analysis described on page 9. They state that "the clustering pattern was different for common variants" but don't test this assertion. They need to show that the difference they observe are unlikely to be due to chance.
- 2) They should test whether the phenotype dependent improvement of PRS when incorporating rare coding variant burden is significant (page 10)

Reviewer #2 (Remarks to the Author):

The authors have made a substantial effort revising the manuscript in response to the reviewers' comments. I recommend acceptance.

Point-by-point response

We thank the reviewers for the positive feedback. Please see our response below. We also highlighted all changes in the manuscript.

REVIEWERS' COMMENTS

Reviewer #1 (Remarks to the Author):

This paper is an exome analysis of different definitions of depression in the UK biobank and makes three points: there is an excess of damaging coding variants in cases of depression, there are different genetic relationships between definitions across common and rare variants, and the effect of damaging coding variant burden and polygenic risk score on depression risk is additive. These are interesting claims and overall the paper does not oversell the findings (as they say, the association results for the rare variant gene burden tests have to be taken as preliminary not definitive)

I have two suggestions to improve the manuscript:

1) The claim that there are different genetic relationships between definitions across common and rare variants relies on a cluster analysis described on page 9. They state that “the clustering pattern was different for common variants” but don’t test this assertion. They need to show that the difference they observe are unlikely to be due to chance.

We thank the reviewer for the comment. Following reviewer’s comment, we estimated the adjusted Rand Index (ARI) with 95% confidence interval (CI) between the clustering results (N=2) for depression definitions based on rare PTV and missense variant and common variant genetic correlations (r_g). ARI was 1 (95% CI: 0.52 to 1.47) between rare PTV and missense variant r_g clusters, which suggests perfect concordance between the two clusterings. The ARI was -0.167 (95% CI: -0.68 to 0.35) between PTV and common variant r_g clusters, which suggests the poor concordance of the two clustering results (as ARI of 0 represents concordance between two randomly clustering and negative value represents especially discordant clusterings). The ARI between rare missense variant and common variant r_g clusters showed the same poor concordance. We revised the Results (page 9) and Methods (page 22) to include these new results.

2) They should test whether the phenotype dependent improvement of PRS when incorporating rare coding variant burden is significant (page 10)

We thank the reviewer for the comment. We have followed the reviewer’s comment to revise the Results (page 10), Methods (page 24), and Supplementary Table 11 to include R^2 comparisons and likelihood ratio tests to compare the PRS models with and without rare coding variant burden and showed different levels of R^2 improvement and model comparison p-values.

Reviewer #2 (Remarks to the Author):

The authors have made a substantial effort revising the manuscript in response to the reviewers' comments. I recommend acceptance.

We thank the reviewer for the feedback.